# Evaluation of Plasma Lipocalin-2 as a Predictor of Etiology and Severity in Adult Patients with Community-Acquired Pneumonia

**DOI:** 10.3390/microorganisms11051160

**Published:** 2023-04-28

**Authors:** Lucía Boix-Palop, Andrea Vergara, Emma Padilla, Diego Martínez, Ana Blanco, Josefa Pérez, Esther Calbo, Jordi Vila, Climent Casals-Pascual

**Affiliations:** 1Infectious Diseases Department, Hospital Universitari Mútua de Terrassa, 08221 Terrassa, Spain; 2School of Medicine, University of Barcelona, 08908 Barcelona, Spain; avergara@clinic.cat (A.V.);; 3School of Medicine, Universitat Internacional de Catalunya, 08195 Barcelona, Spain; 4Biomedical Diagnostic Center (CDB), Department of Clinical Microbiology, Hospital Clinic of Barcelona, 08036 Barcelona, Spain; 5Instituto de Salud Global (ISGlobal), 08036 Barcelona, Spain; 6Department of Clinical Microbiology, Catlab, 08232 Viladecavalls, Spain; 7CIBER de Enfermedades Infecciosas (CIBERINFEC), ISCIII, 28006 Madrid, Spain

**Keywords:** lipocalin-2, community-acquired pneumonia, severity, biomarker, pneumococcal pneumonia

## Abstract

The aim of this study was to evaluate the diagnostic performance of plasma Lipocalin-2 (LCN2) concentration in adult patients with community-acquired pneumonia (CAP) to determine its etiology, severity and prognosis. A prospective observational study involving adults with CAP from November 2015 to May 2017 was conducted. Plasma LCN2 concentration was measured upon admission by a modified enzyme immunoassay coupled with chemiluminescence (Architect, Abbott Laboratories). The diagnostic performance of LCN2, C-reactive protein (CRP) and white blood cell to predict bacterial CAP was assessed. A total of 130 patients with CAP were included: 71 (54.6%) bacterial CAP, 42 (32.3%) unknown origin CAP and 17 (13.1%) viral CAP. LCN2 was higher in bacterial CAP than in non-bacterial CAP (122.0 vs. 89.7 ng/mL, respectively) (*p* = 0.03) with a limited ability to distinguish bacterial and non-bacterial CAP (AUROC: 0.62 [95% CI 0.52–0.72]). The LCN2 cutoff ≥ 204 ng/mL predicted the presence of pneumococcal bacteremia with an AUROC of 0.74 (sensitivity 70%, specificity 79.1%). Regarding severity, as defined by CURB-65 and PSI scores, there was a significant linear trend in the mean concentration of LCN2, exhibiting a shift from the low-risk to the intermediate-risk and high-risk group (*p* < 0.001 and 0.001, respectively). LCN2 concentration was associated with severity in adult patients with CAP. However, its utility as a biomarker to discriminate viral and bacterial etiology in CAP is limited.

## 1. Introduction

Community-acquired pneumonia (CAP) is a lower respiratory tract infection with high morbidity and mortality worldwide [1]. In clinical practice, the presence of symptoms typically associated with pneumonia, namely cough, sputum production, dyspnea, fever and pleuritic chest pain, are not always present, particularly in elderly patients or patients with comorbidities [2]. Approximately 10% of hospitalized patients with CAP require admission to the intensive care unit (ICU) and a significant proportion of patients discharged from the hospital are readmitted within one month [3]. Early diagnosis and treatment of CAP is crucial for the correct clinical management, as severe CAP can be life-threatening. Furthermore, the differentiation between viral and bacterial CAP is essential for the rational administration of antibiotics, de-escalation of empirically administered antibiotics and, ultimately, a reduction in the appearance of antibiotic resistance.

The identification and validation of novel biomarkers to improve the diagnosis of CAP is an emerging and promising area of research in infectious diseases. The currently used biomarkers that may aid in the distinction between bacterial and viral infection are mainly white blood cells (WBC) and C-reactive protein (CRP). However, they typically show clinical performances of 70–80% sensitivity and specificity, which means that a large proportion of infections is misdiagnosed [4,5]. Other inflammatory mediators, such as interleukin (IL)-1β, IL-6, tumor necrosis factor (TNF)-α and IL-8 have also been found to be elevated in response to infection. However, these pro-inflammatory cytokines have very short half-lives and low specificity, presenting a limited diagnostic performance [6].

New biomarkers have been tested for different purposes: to distinguish between pneumonia of viral and bacterial etiology [7], to distinguish between severe and non-severe CAP [3,8,9] and to detect treatment failure [3].

Lipocalin-2 (LCN2), also known as neutrophil gelatinase-associated lipocalin, is a member of the lipocalin family of binding proteins, a large group of small extracellular proteins [10,11]. LCN2 participates in the innate immune response using an iron-depletion strategy [10,12] by interfering with iron acquisition via microorganisms that use sequestering iron-loaded bacterial siderophores, thus limiting their growth. LCN2 is constitutively present in neutrophils and in other tissues, such as colon, uterus, trachea, lung, stomach, prostate and salivary gland [13]. LCN2 concentration increases in kidney tubular cells, intestinal and pulmonary epithelial cells, stomach cells and hepatic cells in response to a variety of stimuli, which include infection or ischemia [12]. Previous studies have reported that LCN2 has an optimal diagnostic performance in discriminating viral and bacterial infections, suggesting that LCN2 is a potential biomarker for clinical diagnosis [4,5,14,15,16,17]. However, the capacity of LCN2 levels to define etiology of CAP in children remains controversial [7,18] and there is less information regarding its clinical performance in adult patients with CAP.

Several studies found that plasma LCN2 concentration is a valuable biological marker in the assessment of the severity and prediction of the prognosis of patients with CAP [7,19,20,21,22,23,24].

The aim of this study was to evaluate the utility of plasma LCN2 concentration to determine the etiology (bacterial vs. non-bacterial), severity and prognosis of adult patients with CAP in the emergency department.

## 2. Materials and Methods

### 2.1. Study Design

A prospective observational cohort study was conducted in a 400-bed teaching hospital in Barcelona, Spain. Consecutive adult patients with CAP who were admitted to the emergency department (ED) from November 2015 to May 2017 were included.

CAP was defined as an acute illness (symptoms lasting for ≤7 days) with the presence of a new infiltrate on a chest radiography associated with two or more of the following signs and symptoms: fever or hypothermia, dyspnea, cough, sputum production, pleuritic chest pain, altered breath sounds on auscultation. The exclusion criteria were: age < 18 years, nosocomial or bronchoaspiratory pneumonia, antibiotic treatment started at the ED > 4 h before the potential inclusion in the study, hospital admission during the previous 14 days or previous inclusion in the study. Pneumococcal CAP (P-CAP) was defined as described in a recently published article by the group [25] (Appendix A). Invasive pneumococcal disease (IPD) was defined as the presence of *S. pneumoniae* in blood cultures, pleural fluid or cerebrospinal fluid. The study was conducted in accordance with the latest version of the Declaration of Helsinki and was approved by the hospital’s Ethics and Research Committee (record 11/15). Written informed consent obtained within 72 h after admission was required for data collection.

### 2.2. Evaluation at Admission and Follow Up

Demographic and clinical variables were prospectively collected as described in the Appendix A. Before starting antibiotic therapy, microbiological studies, chemistry and hematological tests, arterial blood gas sampling and chest radiography were performed (Appendix A). The pneumonia severity index (PSI) [26] and the CURB-65 score [27] were used to grade the severity of the CAP.

### 2.3. LCN2 Quantification

Plasma LCN2 concentration was determined retrospectively on biological samples obtained at admission using a modified coupled enzyme immunoassay with chemiluminescence (Architect Urine NGAL; Abbott Laboratories, Madrid, Spain) and the ARCHITECT i1000SR equipment (Abbott Laboratories, Madrid, Spain) following manufacturer’s instructions. The assay includes a microparticle reagent prepared by covalently attaching an anti-LCN2 antibody to paramagnetic particles and a conjugate reagent prepared by labeling a second anti-LCN2 antibody with acridinium. The assay is completely automated; it allows an analysis of the individual or sets of samples and it provides the results in 30–35 min.

### 2.4. Statistical Analysis

Categorical variables were presented using counts and percentages and continuous variables as medians and interquartile range (IQR). Comparative analyses were performed with the chi-square test or Fisher exact test for categorical variables and Student’s *t*-test, Wilcoxon rank sum test or Kruskall–Wallis test for continuous variables, as appropriate. Relationship between continuous variables was evaluated with Spearman correlation coefficient. Linear trend across ordered groups was evaluated.

The diagnostic performance of plasma LCN2, CRP and WBC count was evaluated by receiver operating characteristic (ROC) curves and the area under the ROC curve (AUC). The optimal cutoff values for the different biomarkers were obtained based on the highest sensitivity and specificity using the roctab function in STATA. Comparison of ROC curves were performed by DeLong’s test. Multivariate logistic regression analyses were performed to evaluate the association of plasma LCN2 levels with the etiology of CAP and with severity based on the CURB-65 score and PSI. Considering that the statistical power of subgroup analysis may be limited by the low number of events, multivariate analyses were not performed in ICU and IPD patients. The variables chosen for inclusion in the multivariate analysis were selected by clinical criteria and those with a univariate *p* value < 0.05. A best subset regression procedure was used to identify the most suitable and parsimonious multivariate model based on the Akaike information criterion [28]. Differences were considered statistically significant at the two-sided *p* < 0.05 level. Statistical analyses were performed using the R Statistical Software (version 4.0.3), RStudio Software (version 1.3.1093) and STATA RELEASE 16 software (StataCorp LP, College Station, TX, USA).

## 3. Results

### 3.1. Clinical and Microbiological Features of the Study Population

A total of 133 episodes of CAP were analyzed during the study period. LCN2 was available in 130 of them, which were finally included in the study: 71 (54.6%) bacterial CAP (including co-infections with other bacteria or viruses), 42 (32.3%) CAP of unknown origin; and 17 (13.1%) viral CAP (including co-infections with another respiratory virus). Among bacterial CAP, 83.3% (60/71) of the cases were P-CAP, with viral co-infection in 46.7% (28/60) cases and bacterial co-infection in 3.3% (2/60) cases. Ten patients had IPD, presenting pneumococcal bacteremia. Table 1 shows the CAP etiology in detail. Patient characteristics and comparison among bacterial, unknown origin CAP and viral CAP are shown in Table 2.

### 3.2. Plasma LCN2 and CAP Etiology

Plasma LCN2 concentration was higher in bacterial CAP (122.0 ng/mL), followed by viral CAP (112.3 ng/mL) and unknown origin CAP (88.2 ng/mL), without reaching statistical significance (*p* = 0.09) (Figure 1). By grouping viral and unknown origin CAP, the concentration of LCN2 was significantly higher in bacterial CAP (122.0 vs. 89.7 ng/mL, respectively) (*p* = 0.03) (Figure 1). The diagnostic performance of LCN2 to predict bacterial CAP is described in Table 3. LCN2 concentration was not associated with bacterial CAP in the multivariate analysis (Table 4). Plasma LCN2 concentration among patients with bacterial, unknown etiology and viral CAP, excluding patients with bacterial and viral coinfection, is shown in Appendix A.

Plasma CRP at admission in the different groups of CAP is shown in Figure 1. The diagnostic performance of CRP is shown in Table 3.

A positive weak correlation was found upon patient admission between plasma LCN2 and CRP (R = 0.31, *p* < 0.01), as well as between plasma LCN2 and WBC count (R = 0.31, *p* < 0.01) (Appendix A). The sensitivity and specificity of a single biomarker to discriminate bacterial from viral CAP were not suitable for clinical use and there was no statistically significant difference between the AUROC of LCN2, CRP and WBC to distinguish bacterial and non-bacterial CAP (Appendix A). The combination of LCN2 and CRP or LCN2 and WBC slightly improve the AUROC of LCN2 alone for the discrimination of bacterial CAP, without statistical significance (Appendix A).

### 3.3. Plasma LCN2 and Pneumococcal Pneumonia

Plasma LCN2 concentration was higher in P-CAP (108.8 ng/mL) than in non-bacterial CAP (89.7 ng/mL), but these differences were not statistically significant (*p* = 0.12). The results were similar regardless of the days of symptoms, the intake of antibiotics 24 h prior to the admission or the presence of renal failure.

In P-CAP patients, plasma LCN2 concentration was significantly higher in patients with bacteremia than in patients with negative blood cultures (285.6 ng/mL (122.0–393.6) vs. 94.1 ng/mL (57.2–166.6), respectively; *p* = 0.005) and in patients with positive PCR- *lytA* in blood compared with those with negative PCR (273.9 ng/mL (169.0–368.1) vs. 96.4 ng/mL (62.5–202.5), respectively; *p* = 0.04) (Figure 2).

LCN2 concentration was associated with pneumococcal bacteremia in the univariate analysis: OR 1.03 (95% CI 1.01–1.05, *p* = 0.026). The diagnostic performance of the biomarkers to predict P-CAP and IPD is shown in Table 3.

### 3.4. Plasma LCN2 as a Biomarker of Severity CAP

Patients were stratified into three risk groups based on the CURB-65 score (0/1; 2; and 3) and based on the PSI (I/II/III: ≤90 points; IV: 90–130 points; V: ≥131). The high-risk group included 16.3% and 14.7% of the CAP patients according to CURB-65 and PSI, respectively. The median (IQR) concentrations of LCN2, CRP, WBC count and lymphocytes count in each CURB-65 and PSI risk group are presented in Table 5.

There was no statistical difference in the LCN2 concentration in patients with diabetes or renal failure, and there was no association between LCN2 concentration and age.

The multivariate analysis showed that plasma LCN2 was independently associated with severity (CURB-65 and PSI) (Table 4).

### 3.5. Plasma LCN2 and Clinical Outcome

Plasma LCN2 concentration at ED admission was higher in patients admitted to the ICU than in patients admitted to the general ward without statistical significance: 150.6 ng/mL (105.6–430.4) vs. 98.1 ng/mL (60.3–184.6), respectively (*p* = 0.06). However, excluding patients that have taken antibiotics in the last 24 h before admission, the difference was statistically significant: 246.7 ng/mL (134.2–548.0) vs. 98.4 ng/mL (61.5–180.2), respectively (*p* = 0.03). We did not find statistically significant differences for CRP and WBC.

LCN2 concentration was not associated with ICU admission in the univariate analysis: OR 1.002 (95% CI 0.99–1.005, *p* = 0.069). However, there was an association between LCN2 concentration and ICU admission in patients without antibiotics in the last 24 h: OR 1.03 (95% CI 1.01–1.05, *p* = 0.04). The optimal LCN2 cutoff concentration for predicting ICU admission was 120.2 ng/mL, with an AUROC of 0.73 (95% CI 0.52–0.94), (sensitivity 87.5%, specificity 59.6%).

## 4. Discussion

In the present prospective cohort study of 130 adult patients with CAP, plasma LCN2 concentration was associated with the severity of CAP, measured by PSI and CURB-65, and seems to be associated with ICU admission and IPD. However, the ability of LCN2 to predict bacterial CAP was similar to other biomarkers as CRP. The use of a chemiluminescence technique allowed us to determine LCN2 concentration easily and rapidly.

LCN2 is stored in the human neutrophils and readily released upon stimulation. It may, however, also be produced by epithelial cells after exposure of the cells to cytokines such us TNF-α [29]. LCN2 exists as a monomer or heterodimer, which is produced by both neutrophils and epithelial cells, and in dimeric form that is exclusively released by neutrophils [4,29]. It has traditionally been used as a marker of acute kidney failure [30], but it has been also proposed as a biomarker of infection [12], including respiratory tract infections (RTI) [7,19,20,21,22,23].

To reduce unnecessary antibiotic treatment, early identification of CAP etiology is critical. In our study, the LCN2 concentration was significantly higher in bacterial CAP than in the group of non-bacterial CAP. However, there was no difference between bacterial and viral CAP. The cutoff value to differentiate between bacterial and non-bacterial CAP was similar to those previously described, with values between 125.9–167.0 ng/mL [4,7,16,31], but it showed limited clinical performance, and it was comparable to other biomarkers as CRP.

Some studies have demonstrated that LCN2 in serum [4,16,31] or measured after the activation of whole blood [14,15,32] could distinguish between acute infections caused by bacteria or virus. Moreover, LCN2 was superior to other biomarkers, such WBC, CRP, procalcitonin or CD64 expression on neutrophils in the distinction between bacterial and viral infection [4,15,31,32].

These findings have been also described in patients with RTI, although there is less information. Previous studies showed significantly higher levels of LCN2 in children with probable bacterial CAP [7] based on clinical and radiological findings with microbiological confirmation in only 6% of cases. However, these results were not confirmed by other studies [18,24] in which a more comprehensive etiological characterization was made. In adults, the diagnostic performance of LCN2 to distinguish bacterial and viral etiology in RTI was optimal, with an AUC > 0.80, and was superior to that of CRP [14,31]. However, these studies did not distinguish between upper and lower RTI, and there is no specific information regarding patients with CAP.

The diagnostic performance of the LCN2 in our study was lower, with an AUC of 0.62. The difference could be explained because all the patients in our cohort were included, following a strict definition of CAP with clinical and radiological criteria. Moreover, etiology was identified in more than two thirds of patients after an exhaustive microbiological study. In contrast, in previous studies, the presence of radiological infiltrates was considered a criterion of possible bacterial etiology [7]; bacterial etiology of the infection was based on a clinical diagnosis without microbiological confirmation [14,15,32] or when only bacterial CAP was described [5,32]. On the other hand, etiology could not be identified in one third of our cohort patients. In this group, some bacterial infections could have been misidentified. This fact added to the ability of viral CAP to cause significant inflammation, which could lead to the release of LCN2 by the pulmonary epithelial cells [4,5] and which could potentially have affected the diagnostic performance of LCN2. In addition, the low severity of our cohort, with only one death, may also have had an influence due to a possible selection bias [14,31].

Regarding pneumococcal pneumonia, the LCN2 cutoff ≥ 204 ng/mL predicted the presence of bacteremia with an AUROC of 0.74 (sensitivity 70%, specificity 79.1%). LCN2 was associated with IPD in the univariate analysis, but we could not further evaluate the association in adjusted models due to the limited sample size. The LCN2 concentration was significantly higher in patients with bacteremia, as well as with *lyt*A-PCR positivity and bacterial load in blood. Unfortunately, this was not observed with less invasive infections, such as those diagnosed using nasopharyngeal swab *lyt*A-PCR or urine antigen detection.

As reported in previous studies [7,19,21,22], we found a relationship between higher LCN2 concentration at ED admission and more severe CAP with both PSI and CURB-65 scores. We could not, however, confirm this observation for CRP or WBC count.

The prognostic value of LCN2 has been previously evaluated. Indeed, Min et al. [21] found that plasma LCN2 was a useful biomarker for predicting ICU admission and mortality in hospitalized patients with pneumonia. Similarly, we found that LCN2 concentration was significantly higher in patients who required ICU admission and an association between LCN2 and ICU admission in the univariate analysis. We could not analyze the usefulness of LCN2 as a biomarker of in-hospital or 30-day mortality because only one patient died in the hospital and none of them during the 30 days of follow-up.

In our cohort, there was no difference in the LCN2 concentration in patients with diabetes or renal failure, and there was no association between LCN2 concentration and age. This is in contrast with a previously published article in which LCN2 levels were higher in patients with diabetes than in non-diabetic patients [33], as well as some studies that have reported a correlation between LCN2 and age [34], but which were conducted on patients without an acute infection.

The major strength of this study is that the patients included were part of a well characterized prospective cohort, with an etiological diagnosis established in almost 70% of CAP episodes [25]. Moreover, this is the first study to our knowledge in which a chemiluminescence technique is used to evaluate the LCN2 as a biomarker of CAP. In most papers, LCN2 concentration is measured with enzyme-linked immunosorbent assay (ELISA), which is neither easy nor rapid. The assay used in the present study is completely automated, allowing individualized sample testing with the absolute quantification of LCN2 and a turnaround time of 30 min, making this method suitable for point-of-care use. It is a simple and accessible test that allows us to quickly identify patients at high risk of presenting severe CAP, and so a close clinical follow-up can be carried out.

The main limitations of the present study are: (a) the limited number of patients and the observational design of the study that precluded any conclusion about the effectiveness of using LCN2 as a biomarker for the management of CAP; (b) the concentration of LCN2 was measured at a single time point, making it impossible to evaluate the kinetics of this biomarker over the course of infection; (c) the time between the start of the CAP symptoms and the time of blood samples was not constant; (d) the proportion of patients with severe CAP was relatively small and mortality was extremely low, perhaps due to a selection bias; (e) the statistical power of subgroup analysis may be limited by the low number of events in every subgroup; and (f) the assay used to determine plasma LCN2 concentration detected the monomeric form, which is also secreted by the pulmonary epithelial cells. Finally, our study was conducted in a specific geographical area and the results cannot be extrapolated to other settings.

## 5. Conclusions

LCN2 concentration was associated with severity in adult patients with CAP and seems to be associated with ICU admission and IPD. However, the ability of LCN2 as a biomarker to discriminate CAP etiology was limited. The technique used in this study make it possible to determine LCN2 concentration in a reasonable time to inform clinical decisions. Interventional studies should be conducted to further evaluate the potential clinical and cost-effectiveness of this biomarker in CAP.

## Figures and Tables

**Figure 1 microorganisms-11-01160-f001:**
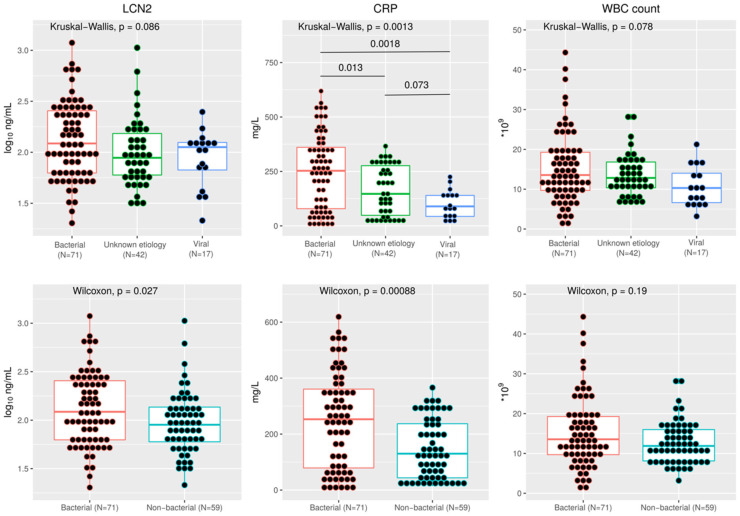
Concentration of LCN2, CRP and WBC count among patients with bacterial, unknown etiology and viral CAP (**above**) and between bacterial and non-bacterial CAP (**below**).

**Figure 2 microorganisms-11-01160-f002:**
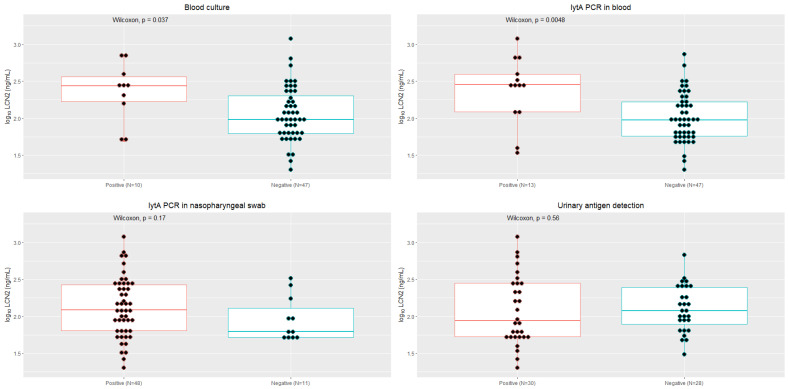
Concentration of LCN2 in pneumococcal community-acquired pneumonia episodes depending on the result obtained from blood culture (positive or negative for *S. pneumoniae*), *lytA* PCR in blood, *lytA* PCR in nasopharyngeal swab and urinary antigen detection.

**Table 1 microorganisms-11-01160-t001:** Etiology of community-acquired pneumonia (CAP) episodes included in the study (N = 130).

Microorganism	N (%)
**Bacterial CAP**	**71 (54.6)**
*Streptococcus pneumoniae*	60 (46.2)
- *S. pneumoniae* co-infection *	30 (23.1)
*Mycoplasma pneumoniae*	2 (1.54)
*Legionella pneumophila*	2 (1.54)
*Chlamydophila pneumoniae* + Human rhinovirus	2 (1.54)
*Streptococcus pyogenes*	1 (0.77)
*Staphylococcus aureus* + Influenza virus	1 (0.77)
*C. pneumoniae * + Adenovirus	1 (0.77)
*Streptococcus viridans* + Respiratory syncytial virus	1 (0.77)
*Legionella pneumophila* + Adenovirus	1 (0.77)
**Unknown origin CAP**	**42 (32.3)**
**Viral CAP**	**17 (13.1)**
Human rhinovirus	5 (3.84)
Influenza virus	3 (2.31)
Metapneumovirus	2 (1.54)
Influenza virus + Adenovirus	2 (1.54)
Coronavirus	1 (0.77)
Parainfluenza virus	1 (0.77)
Respiratory syncytial virus	1 (0.77)
Adenovirus	1 (0.77)
Human rhinovirus + Adenovirus	1 (0.77)

* *S. pneumoniae* co-infection: viral: 15 human rhinovirus, 5 influenza virus, 3 coronavirus, 2 respiratory syncytial virus, 1 adenovirus, 1 metapneumovirus, 1 parainfluenza virus, 1 *Haemophilus influenza*, 1 *Moraxella catharralis.*

**Table 2 microorganisms-11-01160-t002:** General characteristics of the 130 patients included and the comparison between those with bacterial CAP, unknown origin CAP and viral CAP.

	Bacterial CAP (N = 71)	Unknown Origin CAP (N = 42)	Viral CAP (N = 17)	*p*
**Demographic data**				
Age (years)	68.0 (53.5–79.0)	72.5 (59.0–82.0)	82.0 (76.0–83.0)	**0.02** ^A^
Gender, male	50 (70.4)	28 (66.7)	6 (35.3)	**0.03** ^A^
BMI (kg/m^2^)	25.6 (22.3–29.1)	26.9 (23.9–29.8)	27.9 (22.7–30.8)	0.3
Caucasian race	70 (98.6)	41 (97.6)	16 (94.1)	0.3
Current smoker	18 (25.3)	12 (28.6)	0	**0.03** ^A,C^
Ex-smoker	23 (32.4)	17 (40.5)	6 (35.3)	0.7
**Comorbid conditions**				
Charlson index ≥ 4	10 (14.1)	4 (9.5)	2 (11.8)	0.9
Chronic kidney disease	8 (11.3)	5 (9.5)	3 (17.6)	0.4
Diabetes	54 (76.1)	33 (78.6)	14 (82.4)	0.62
Immunosuppression ^1^	4 (5.6)	0	2 (11.8)	0.09
Prior antibiotic treatment ^2^	9 (12.7)	7 (16.7)	4 (23.5)	0.5
Prehospital treatment ^3^	11 (15.5)	4 (9.5)	3 (17.6)	0.6
**Clinical features on presentation**				
Time symptom onset—ED visit (hours)	48 (24–108)	84 (48–138)	72 (48–96)	0.1
Fever (≥38 °C)	35 (49.3)	12 (28.6)	6 (35.3)	0.09
Dyspnea	50 (70.4)	32 (76.2)	12 (70.6)	0.8
Tachypnea (≥20 rpm)	45 (69.2)	24 (64.9)	8 (61.5)	0.8
Cough	63 (88.7)	35 (83.3)	16 (94.1)	0.6
Purulent sputum	30 (42.3)	19 (45.2)	9 (52.9)	0.8
Pleuritical chest pain	30 (42.3)	10 (23.8)	3 (17.6)	0.05
Septic shock	16 (22.9)	3 (7.1)	2 (11.8)	0.08
Respiratory failure	38 (55.1)	23 (54.8)	13 (76.5)	0.3
PaO_2_/FiO_2_	290 (257.5–345.5)	300 (252–362)	271 (243–300)	0.1
PSI ≥ 4 *	38 (54.3)	14 (33.3)	11 (64.7)	0.1
CURB-65 score ≥ 3 *	11 (15.7)	7 (16.7)	3 (17.6)	0.6
**Laboratory findings ***				
LCN2 (ng/mL)	122.0 (62.6–255.3)	88.2 (59.8–153.0)	112.3 (67.0–124.8)	0.08
CRP (mg/L)	253.0 (79.0–360.7)	147.0 (48.5–276.5)	89.5 (43.5–140.0)	**0.001** ^A,B^
WBC count (×10^9^/L)	13.5 (9.7–19.3)	12.8 (10.4–16.8)	10.3 (6.6–14.0)	0.08
Lymphocytes count (×10^9^/L)	1.0 (0.7–1.4)	1.2 (0.8–1.6)	1.2 (0.7–1.7)	0.4
**Evolution and Outcome**				
Time to clinical stability (days)	2 (1–4)	2 (1–4)	2 (1–3)	0.6
ICU admission	7 (9.9)	3 (7.1)	1 (5.9)	0.9
Mechanical Ventilation	3 (4.2)	2 (4.8)	1 (5.9)	1
Length of hospital stay (days)	6.0 (4.0–8.8)	5.5 (1.2–9.0)	6.0 (5.0–9.0)	0.7
In-hospital mortality	0	0	1 (5.9)	0.1

Values are n (%) or median (interquartile range). BMI: Body mass index; ED: emergency department; FIO_2_: fraction of inspired oxygen; ICU: intensive care unit; PSI: pneumonia severity index; CURB-65: confusion, urea, respiration, blood pressure, age ≥ 65 years; LCN2: lipocalin-2; CRP: C-reactive protein; WBC: white blood cells. ^A^: Statistically significant between bacterial and viral. ^B^: Statistically significant between bacterial and unknown origin. ^C^: Statistically significant between unknown origin and viral. ^1^ Immunosuppression: Chronic corticosteroid therapy, severe neutropenia, solid or hematopoietic organ transplantation, acquired immunodeficiency syndrome and use of chemotherapy, immunosuppressive agents or biological drugs. ^2^ Prior antibiotic treatment: Intake of antibiotic 3 months before hospitalization. ^3^ Prehospital antibiotic treatment: Oral intake of antibiotic > 24 h prior to hospitalization for the same episode of acute disease. *: at admission.

**Table 3 microorganisms-11-01160-t003:** Diagnostic performance of the biomarkers to predict the different CAP etiologies, odds ratio and 95% confidence interval according to the biomarker cutoff.

Biomarker	Cutoff Value	Sensitivity (%)	Specificity (%)	PPV (%)	NPV (%)	AUC (95% CI)	OR (95% CI)
**Bacterial CAP (n = 71)**							
-LCN2 (ng/L) -CRP (mg/L) -WBC (mm^3^)	138.9 204 13.5	77.9 61.8 50.7	47.8 71.2 62.7	72.3 71.2 61.4	55.4 61.8 52.1	0.62 (0.52–0.72) 0.67 (0.58–0.77) 0.57 (0.47–0.67)	3.25 (1.50–7.04) 4.00 (1.89–8.42) 1.73 (0.85–3.51)
***S. pneumoniae* CAP (n = 60)**							
-LCN2 (ng/L) -CRP (mg/L) -WBC (mm^3^)	138.9 204 13.5	43.4 62.1 52.5	70.0 66.7 62.3	55.3 61.0 54.4	59.0 67.6 60.6	0.54 (0.44–0.65) 0.64 (0.54–0.74) 0.57 (0.46–0.67)	1.78 (0.87–3.68) 3.27 (1.58–6.79) 1.83 (0.90–3.71)
**IPD (n = 10)**							
-LCN2 (ng/L) -CRP (mg/L) -WBC (mm^3^)	204 318 17.0	70.0 80.0 60.0	79.1 83.2 73.7	22.6 29.6 16.7	96.8 97.9 95.5	0.74 (0.53–0.94) 0.82 (0.71–0.93) 0.56 (0.30–0.82)	8.85 (2.13–36.8) 21.1 (4.14–107.7) 4.2 (1.11–15.9)
**Viral CAP (n = 17) ***							
-LCN2 (ng/L) -CRP (mg/L) -WBC (mm^3^)	98.1 88.2 12.8	58.8 58.8 35.3	48.7 30.9 48.6	14.7 11.6 9.5	88.7 82.9 83.1	0.41 (0.33–0.51) 0.29 (0.20–0.39) 0.33 (0.20–0.47)	1.34 (0.48–3.81) 0.64 (0.22–1.82) 0.52 (0.18–1.49)

CAP: Community-acquired pneumonia; PPV: positive predictive value; NPV: negative predictive value; AUC: area under the ROC curve; CI: confidence interval; IPD: invasive pneumococcal diseases. Other abbreviations as in Table 1. * Inverse relationship between viral CAP and biomarkers.

**Table 4 microorganisms-11-01160-t004:** Univariate and multivariate logistic regression for the association of LCN2 plasma concentration with bacterial CAP, high risk CURB65 and high risk PSI.0.009 0.72 (0.59–0.87).

	Bacterial CAP		CURB-65 ≥ 3		PSI ≥ 4	
	OR (95% CI)	*p*	OR (95% CI)	*p*	OR (95% CI)	*p*
Univariate						
Age (years)	0.97 (0.95–0.99)	0.03	1.04 (1.00–1.08)	0.03	1.04 (1.02–1.07)	0.001
Gender, male	0.57 (0.28–1.18)	0.13	0.88 (0.33–2.38)	0.81	0.54 (0.26–1.13)	0.10
Current smoker	1.33 (0.58–3.05)	0.50	0.30 (0.66–1.37)	0.12	0.75 (0.33–1.71)	0.49
Charlson index ≥ 4	1.45 (0.49–4.25)	0.50	7.69 (2.47–24.0)	<0.001	9.14 (1.98–42.1)	0.005
Renal failure	0.94 (0.32–2.71)	0.92	6.25 (1.96–19.9)	0.002	18.6 (2.36–146.1)	0.005
Prehospital treatment	1.36 (0.49–3.77)	0.55	0.65 (1.96–19.9)	0.59	0.39 (0.13–1.17)	0.09
Time from symptom onset to ED visit (days)	0.85 (0.71–1.01)	0.06	0.66 (0.48–0.90)	0.009	0.72 (0.59–0.87)	0.001
Septic shock	3.20 (1.09–9.35)	0.03	8.02 (2.78–23.1)	<0.001	4.15 (1.42–12.2)	0.009
Respiratory failure	0.80 (0.40–1.62)	0.54	2.81 (0.96–8.21)	0.06	7.02 (3.17–15.5)	<0.001
LCN2 (10 ng/mL)	1.02 (0.99–1.05)	0.07	1.06 (1.03–1.09)	<0.001	1.04 (1.01–1.07)	0.01
CRP (mg/dL)	1.05 (1.02–1.08)	<0.001	1.01 (0.98–1.04)	0.42	1.01 (0.98–1.03)	0.66
Best predictive model						
Age (years)	0.97 (0.94–0.99)	0.009	1.07 (1.02–1.13)	0.008	1.04 (1.01–1.08)	0.008
Charlson index ≥ 4			6.77 (1.51–30.4)	0.013	6.47 (1.03–40.8)	0.047
Time from symptom onset to ED visit (days)	0.71 (0.57–0.88)	0.002	0.70 (0.44–1.12)	0.14	0.79 (0.63–0.99)	0.047
Septic shock			8.74 (1.77–43.3)	0.008		
Respiratory failure					6.07 (2.44–15.1)	<0.001
LCN2 (10 ng/mL)	1.01 (0.98–1.04)	0.42	1.04 (1.01–1.09)	0.029	1.04 (1.01–1.07)	0.025
CRP (mg/dL)	1.06 (1.02–1.09)	0.0001				

OR: Odds ratio; CI: confidence interval; other abbreviations as in Table 1.

**Table 5 microorganisms-11-01160-t005:** Biomarker concentration by CURB-65 and PSI classification and linear trend analysis.

	CURB-65				PSI			
	0/1	2	≥3	*p*	I/II/III	IV	V	*p*
**Number (%)**	71 (55.0)	37 (28.7)	21 (16.3)		66 (51.2)	44 (34.1)	19 (14.7)	
**LCN2 (ng/mL)**	88.2 (53.5–144.4)	115.0 (62.0–243.1)	265.0 (138.8–380.2)	<0.001 *	90.9 (54.1–159.6)	107.4 (59.9–176.5)	265.0 (120.3–352.4)	0.001 *
**CRP (mg/L)**	195.0 (67.8–297.8)	143.0 (44.7–326.0)	240.0 (85.5–312.2)	0.44	193.0 (59.6–299.0)	203.0 (76.1–326.0)	93.0 (48.1–289.0)	0.96
**WBC (×10^9^/L)**	11.5 (8.5–16.1)	12.8 (10.3–19.3)	16.6 (10.9–23.2)	0.02	12.7 (8.9–16.1)	13.2 (9.3–19.3)	13.8 (8.4–22.2)	0.23
**Lymphocytes (×10^9^/L)**	1.1 (0.7–1.5)	1.0 (0.5–1.4)	1.3 (0.8–1.7)	0.42	1.1 (0.7–1.5)	1.0 (0.7–1.6)	1.2 (0.7–1.7)	0.52

Values are n (%) or median (interquartile range). LCN2: Lipocalin-2; CRP: C-reactive protein; WBC: white blood cells. Other abbreviations as in Table 1. * *p* < 0.05 also observed in the comparison of the LCN2 concentration between the different groups.

## Data Availability

The original contributions presented in the study are included in the article/Appendix A, further inquiries can be directed to the corresponding authors.

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
