# Peer review of "Evaluation of Plasma Lipocalin-2 as a Predictor of Etiology and Severity in Adult Patients with Community-Acquired Pneumonia"

_microorganisms, 2023, doi:10.3390/microorganisms11051160_

Round 1

Reviewer 1 Report

Dear Editor,

Thank you for inviting me to review this manuscript. Boix-Palop et al. included 130 adult patients with community-acquired pneumonia (CAP) in this observational study. 55% of the patients had bacterial-CAP, 13% viral-CAP, and the etiology was unknown in 32% of patients.

To predict bacterial CAP, the authors measured lipocalin-2 (LCN2), CRP, and white blood cells. LCN2 values were higher in bacterial-CAP than in non-bacterial-CAP. Furthermore, LCN2 was correlated with pneumonia severity according to CURB-65 and PSI scores. Although LCN2 could not distinguish bacterial and non-bacterial-CAP, the authors determined a cut-off for LCN2 to predict the presence of pneumococcal bacteremia.

The subject is interesting, and the manuscript is well-written. I only have some minor comments:

1-      On page 2, lines 51-52, “However, the capacity of LCN2 levels to define etiology of CAP in children remains controversial [7,18]”. I suggest removing this sentence. Because the LCN2 levels are correlated with age (Please see PMID: 34026950). Therefore, this sentence about children seems misleading.

2-      On page 4, lines 103-104, please remove the following sentences related to the template. “This section may be divided by subheadings. It should provide a concise and precise description of the experimental results, their interpretation, as well as the experimental conclusions that can be drawn.”

3-      It has been reported that LCN2 levels are higher in patients with diabetes than in non-diabetic patients (PMID: 28709459). However, there is no information about the proportion of diabetic patients in this manuscript. I suggest adding the proportion of diabetic patients to Table 2 and, if relevant, adjusting for diabetes in the logistic regression model.

4-      Please add to the discussion the role of age and diabetes on LCN2 levels.

Author Response

RESPOND TO REVIEWER 1 COMMENTS

Manuscript Number: microorganisms-2321196
Article Title: Evaluation of Plasma Lipocalin-2 as a Predictor of Etiology and Severity in Adult Patients with Community-Acquired Pneumonia

REVIEWER 1 COMMENTS

Reviewer #1:

Dear Editor,

Thank you for inviting me to review this manuscript. Boix-Palop et al. included 130 adult patients with community-acquired pneumonia (CAP) in this observational study. 55% of the patients had bacterial-CAP, 13% viral-CAP, and the etiology was unknown in 32% of patients. 

To predict bacterial CAP, the authors measured lipocalin-2 (LCN2), CRP, and white blood cells. LCN2 values were higher in bacterial-CAP than in non-bacterial-CAP. Furthermore, LCN2 was correlated with pneumonia severity according to CURB-65 and PSI scores. Although LCN2 could not distinguish bacterial and non-bacterial-CAP, the authors determined a cut-off for LCN2 to predict the presence of pneumococcal bacteremia.

The subject is interesting, and the manuscript is well-written. I only have some minor comments:

1-      On page 2, lines 51-52, “However, the capacity of LCN2 levels to define etiology of CAP in children remains controversial [7,18]”. I suggest removing this sentence. Because the LCN2 levels are correlated with age (Please see PMID: 34026950). Therefore, this sentence about children seems misleading. 

Response:

We agree with the Reviewer 2 that there are some articles published that have found a relation between the plasma LCN2 concentration and age, describing an increasing of the LCN2 with the age. However, in our opinion, it is necessary to emphasize some aspects before generalizing the relationship between age and LCN concentration described in these articles:

  • First, LCN2 is a protein that exits as a monomer or heterodimer, which is produced by both neutrophils and epithelial cells, and as dimeric form, that is exclusively released by neutrophils. When we are evaluating and infection episode, such as CAP, the LCN2 concentration will raise due to the increase of all the forms, but with a predominant role of the dimeric form which is produced by neutrophils. However, in other situations related with chronic inflammation (such as the conditions described in the articles (the one facilitated by the reviewer and others cited on them): Maurizzi et al -PMID: 34026950- osteoporosis and bone metabolism; Bahmani et al -PMID:24452457 - healthy volunteers; Bachorzeswka-Gajewska et al -PMID: 16772710- patients undergoing and elective percutaneous coronary intervention) the LCN2 concentration will be elevated due to the epithelial cells production. There is no description of the type of LCN2 measured in these articles.
  • Second, in most of the articles they did not include children and they have a small sample size (30, 35 patients…)
  • Third, in none of the articles did the patients have an acute infection
  • There is one study with pediatric population 25348707) and they evaluated the role of LCN2 concentration in urine to predict early kidney injury. They found higher LCN2 concentration in children with 15-18 years than in children with 3-5 years, but with values of 12.1 ng/mL (IQR 6.4-27) and 3.6 ng/mL (IQR 1.9-12), which are clearly smaller than the ones reported in our article and in others (also in children) realized in patients with acute infection.
  • Finally, in the article conducted by Bachorzeswka-Gajewska et al, they found a relation between age and LCN2 concentration and also between creatinine levels and LCN2 concentration in the univariate analysis, but in the multivariate analysis serum creatinine was the only predictor of serum LCN2.

It is possible that the secretion of LCN2 produced by epithelial cells is increased by age due to the inflammation that produces senescence, as described by Maurizi et al, but the secretion produced by both neutrophils and epithelial cells in an acute infection would probably be higher. In addition, it is plausible to consider that the release of LCN2 produced during acute infection is greater in those young patients without comorbidities who have an immune system capable of developing a good inflammatory response.

In our study there was no correlation between the age and the LCN2 concentration, as shown by the Pearson correlation analysis: Spearman’s rho = -0.086, p = 0,29.

Moreover, although the value in near 0, it is a negative value, so if there exists a trend to any relation it would be to a higher LCN2 concentration in the younger patients.

We add also a graphic studying this correlation:

*EDAD = age

Considering our results and the previously explained arguments, we consider that we cannot determine the relationship between age and LCN2 in episodes of acute infection. Therefore, if there is no objection, we will keep the introduction sentence and add, as the reviewer suggests, both in the results (see lines 212-213) and in the discussion (see lines 280-283) a comment about the relationship of LCN2 with age.

2-      On page 4, lines 103-104, please remove the following sentences related to the template. “This section may be divided by subheadings. It should provide a concise and precise description of the experimental results, their interpretation, as well as the experimental conclusions that can be drawn.”

Response:

We would like to thank the reviewer for his appreciation. We have now modified this in the revised version of the manuscript. 

3-      It has been reported that LCN2 levels are higher in patients with diabetes than in non-diabetic patients (PMID: 28709459). However, there is no information about the proportion of diabetic patients in this manuscript. I suggest adding the proportion of diabetic patients to Table 2 and, if relevant, adjusting for diabetes in the logistic regression model.

Response:

We would like to thank the reviewer for his appreciation. We have now modified this in the revised version of the manuscript.  See Table 2.

We have also performed the relation between the plasma lipocalin-2 concentration and the presence of diabetes by the chi2 test with these results: LCN2 concentration in patients with diabetes 88.3 (IQR 52-131.9) vs 117.2 (IQR 62.7-221.8) in patients without diabetes, with no statistical significance (p=0.094).

As the results are not relevant, we have not added the variable diabetes in the logistic regression model.

4-      Please add to the discussion the role of age and diabetes on LCN2 levels.

Response:

Done as suggested. We have now included this information in the revised version of the manuscript.  See lines 280-283.

Reviewer 2 Report

The authors reported that evaluate the diagnostic performance of plasma LCN2 concentration in adult patients with CAP to determine the etiology, severity, and prognosis.
Let me address some concerns.
Q1. Do not use abbreviations for abstract, and write the full term.
Q2. What is the difference between references 7,19-24 and this study? It is judged that the research design and the outcome you want to check are similar.
Q3. The authors mentioned LCN2 as a marker that can distinguish between etiology (bacterial vs non-bacterial). Have you ever compared it to procalcitonin? It is a more specific maker than WBC and CRP, and as a predictor of etiology, it is recommended to add a comparison of procalcitonin and sensitivity.
Q4. Do LCN2 and other blood sample results (CRP, WBC, lymphocyte, etc) have the same blood sampling point? Does the method section say before starting antibiotic therapy, is the sampling time the same? This can act as a significant bias in the interpretation of the results.
Q5. It is judged that the interval of the study subjects will not be constant between the start of the CAP clinical symptoms and the time of blood sampling. This is an important limitation of the study of pneumonia, including CAP. Please add this point to the revised manuscript.

Author Response

RESPOND TO REVIEWER 2 COMMENTS

Manuscript Number: microorganisms-2321196
Article Title: Evaluation of Plasma Lipocalin-2 as a Predictor of Etiology and Severity in Adult Patients with Community-Acquired Pneumonia

REVIEWER 2 COMMENTS

Reviewer #2:

The authors reported that evaluate the diagnostic performance of plasma LCN2 concentration in adult patients with CAP to determine the etiology, severity, and prognosis.
Let me address some concerns.
Q1. Do not use abbreviations for abstract, and write the full term.

Response:

We would like to thank the reviewer for his appreciation. We have now modified this in the revised version of the manuscript.

Q2. What is the difference between references 7,19-24 and this study? It is judged that the research design and the outcome you want to check are similar.

Response:

The aim of our study was to evaluate the utility of plasma LCN2 concentration to determine: 1. the etiology (bacterial vs. non-bacterial) and 2. the severity and prognosis of adult patients with CAP, from a region with low prevalence of HIV infection.

We agree with Reviewer 2 that there are some similitudes with the studies with reference nº 7, 19 and 24 as all of them evaluate the relation between LCN2 concentration and the severity or mortality of low respiratory tract infections (LRTI) or CAP. But there are also important differences between these studies and our study.

  • The study published by Huang et al (reference 7) is conducted in African children (from Gambia and Kenia) to identify and validate markers of severity in patients with pneumonia. They found that LCN2 was the best protein biomarker of severe pneumonia and also predictive of “probable bacterial etiology”. But, as we explain in the Discussion, the diagnosis of “probable bacterial etiology” was based on clinical and radiological findings with microbiological confirmation in only 6% of cases while in our study the etiological diagnostic was established in almost 70% of the CAP episodes.
  • The study performed by Sawatsky et al (reference 24) is also conducted in African children (from Uganda) with the aim of describing the host inflammatory biomarkers and outcomes among children hospitalized with respiratory syncytial virus LRTI. They found an association between LCN2 concentration and the severity of the episodes, but they evaluated viral LRTI and they did not evaluate the capacity of LCN2 to predict the etiology of the infection.
  • The study done by Kim et al (reference 19) was conducted in adults, evaluating the relationship between the LCN2 concentration and the severity and mortality of CAP. But they did not evaluate the role of LCN2 in predicting the etiology of CAP.

Q3. The authors mentioned LCN2 as a marker that can distinguish between etiology (bacterial vs non-bacterial). Have you ever compared it to procalcitonin? It is a more specific maker than WBC and CRP, and as a predictor of etiology, it is recommended to add a comparison of procalcitonin and sensitivity.

Response:

We agree with Reviewer 2 that procalcitonin (PCT) is a good biomarker to discriminate bacterial and viral infection.

PCT is not routinely used in our hospital as a blanket indication, but rather in selected cases. Indeed, the application of PCT for patients admitted with suspected CAP; for instance, we do not use PCT to guide decisions on antibiotic initiation when the probability of bacterial CAP is moderate or greater in high risk-populations as immunocompromised patients or in those with severe disease. For this reason, we did not include PCT in our study as a routine biomarker such CRP or WBC.

In addition, we did not add PCT to the analysis due to the following reasons:

  1. The aim of the present study is to evaluate the diagnostic performance of plasma LCN2 concentration by itself in a well characterized prospective cohort, with an etiological diagnosis established in almost 70% of CAP episodes.
  2. The performance of and comparisons among other biomarkers obtained specifically for this study, such as adrenomedullin, SuPAR or PCT, is being analyzed but we still do not have the results, which will be published in another manuscript.
  3. On the other hand, we agree with the Reviewer that PCT has some evidence that demonstrate that a PCT guided strategy may reduce antibiotic initiation and duration (Menéndez R, et al. Cytokine activation patterns and biomarkers are influenced by microorganisms in community-acquired pneumonia. Chest. 2012 Jun;141(6):1537–45, Schuetz P, et al. Procalcitonin to initiate or discontinue antibiotics
    in acute respiratory tract infections. Cochrane Database Syst Rev. 2017;10), but there are as well multiple caveats. In a recent meta-analysis performed to evaluate the ability of biomarkers to identify the etiology of pneumonia PCT did not have diagnostic validity as an independent test to differentiate between viral and bacterial infection: its sensitivity was 44-74% and its specificity 74-93%. Selecting the cut-off point > 0.1 μg/L, the sensitivity and specificity were 74% (Gerntilotti E, et al. Diagnostic accuracy of point-of-care tests in acute community-acquired lower respiratory tract infections. A systematic review and meta-analysis. Clin Microbiol Infect. 2022;28(1):13–22). Besides, the 2019 guideline on community-acquired pneumonia from the American Thoracic Society and Infectious Diseases Society of America recommends against the use of PCT testing to guide initiation of empirical antibiotic therapy for radiologically confirmed pneumonia (strong recommendation, moderate evidence), based on a study of the Centers for Disease Control and Prevention EPIC (Etiology of Pneumonia in the Community). In this cohort, viral and bacterial pathogens were identified in 24%
    and 14% of cases, respectively. Although PCT concentrations were significantly higher in the bacterial infection group, the negative predictive value of a PCT value of less than 0.1 ng/mL was 82.4% (95% CI, 71.2% to 86.9%), so, approximately 1 in 5 patients with microbiologically confirmed bacterial CAP had a negative PCT test result (Self WH, et al. Procalcitonin as a marker of etiology in adults hospitalized with community-acquired pneumonia. Clin Infect Dis. 2017;65:183–190). Therefore, the optimal use of PCT in patients with suspected or confirmed pneumonia remains an area of significant controversy.
  4. Lastly, some studies have compared the diagnostic capacity of PCT and LCN2 to differentiate bacterial and viral infections, showing that LCN has better performance that PCT. In the study published by Fang et al. in 2019 the AUROC of LCN2 was 0.91 while that of PCT was 0.63, and in the study of Venge et al, published in 2017, the AUROC of LCN for URTI was 0.92 (0.82-0.97) while the AUROC of PCT was 0.68 (0.56-0.79) (p < 0.001).

Q4. Do LCN2 and other blood sample results (CRP, WBC, lymphocyte, etc) have the same blood sampling point? Does the method section say before starting antibiotic therapy, is the sampling time the same? This can act as a significant bias in the interpretation of the results.

Response:

For the development of this study a protocol of laboratory requests was performed, so at the moment in which a patient was included in the study all the blood samples for the different analyzes planned for the study were carried out (conventional and molecular microbiological studies, chemistry and hematological tests, plasma biomarkers, arterial blood gas sampling and also chest radiography). These blood samples were ideally taken before starting antibiotic therapy and were always taken before 4h of antibiotic treatment started at the Emergency Department (it was one of the exclusion criteria of the study as described in Materials and methods: study design).

The median time in our complete cohort from the blood extraction to the antibiotic administration was 10 minutes (IQR -44 – 49.5 minutes).

In some exceptional cases these extraction was not at the same time of the conventional chemistry and hematological test, as the last one was taken before including the patient in the study, but we don’t have the exact number of cases.

Q5. It is judged that the interval of the study subjects will not be constant between the start of the CAP clinical symptoms and the time of blood sampling. This is an important limitation of the study of pneumonia, including CAP. Please add this point to the revised manuscript.

Response:

We agree with Reviewer 2 that the interval between the start on clinical symptoms and the time of blood sampling is heterogenous, as it is conditionate by the time from symptoms onset to the Emergency Department visit, and this can be a limitation in the study of the etiology and the prognosis of CAP. As suggested, we have included this issue to the revised manuscript (see Discussion, limitations of the study, line 294).

In our study we have describe the time from symptoms onset to the ED visit, without significant differences between the different etiological groups (see Table 2) and without any correlation between the time from symptoms onset and the LCN2 concentration (Pearson correlation no significant, Spearman’s rho=-0.095, p=0.28).
